# Advancing Near-Infrared Probes for Enhanced Breast Cancer Assessment

**DOI:** 10.3390/s25030983

**Published:** 2025-02-06

**Authors:** Mohammad Pouriayevali, Ryley McWilliams, Avner Bachar, Parmveer Atwal, Ramani Ramaseshan, Farid Golnaraghi

**Affiliations:** 1School of Mechatronic Systems Engineering, Simon Fraser University, Surrey, BC V3T 0A3, Canada; mohammad_pouriayevali@sfu.ca (M.P.); rmcwilli@sfu.ca (R.M.); avner.bachar@ufv.ca (A.B.); ramani_ramaseshan_2@sfu.ca (R.R.); 2Automation and Robotics Technician, University of Fraser Valley, Chilliwack, BC V2R 0N3, Canada; 3Department of Medical Physics, BC Cancer, Abbotsford, BC V2S 0C2, Canada; patwal@bccancer.bc.ca

**Keywords:** biosensor, machine learning, breast cancer, chemotherapy, residual cancer burden

## Abstract

Breast cancer remains a leading cause of cancer-related deaths among women, emphasizing the critical need for early detection and monitoring techniques. Conventional imaging modalities such as mammography, MRI, and ultrasound have face sensitivity, specificity, cost, and patient comfort limitations. This study introduces a handheld Near-Infrared Diffuse Optical Tomography (NIR DOT) probe for breast cancer imaging. The NIRscan probe utilizes multi-wavelength light-emitting diodes (LEDs) and a linear charge-coupled device (CCD) sensor to acquire real-time optical data, reconstructing cross-sectional images of breast tissue based on scattering and absorption coefficients. With wavelengths optimized for the differential optical properties of tissue components, the probe enables functional imaging, distinguishing between healthy and malignant tissues. Clinical evaluations have demonstrated its potential for precise tumor localization and monitoring therapeutic responses, achieving a sensitivity of 94.7% and specificity of 84.2%. By incorporating machine learning algorithms and a modified diffusion equation (MDE), the system enhances the accuracy and speed of image reconstruction, supporting rapid, non-invasive diagnostics. This development represents a significant step forward in portable, cost-effective solutions for breast cancer detection, with potential applications in low-resource settings and diverse clinical environments.

## 1. Introduction

Breast cancer remains the most prevalent cancer among women and is the second leading cause of cancer-related deaths [1]. In 2024, it was estimated that approximately 42,250 women in the US would die of breast cancer [1]. In recent years, a variety of imaging modalities have been developed to facilitate early detection and reduce breast cancer mortality rates [2,3]. Despite X-ray mammography being the primary screening modality, it has notable limitations, particularly for younger women and those with dense breast tissue, reducing its sensitivity [2,3]. Additionally, exposure to ionizing radiation can pose long-term health risks [4]. Ultrasound (US) has been utilized as an alternative for imaging dense breast tissues, but its effectiveness is hindered by speckle artifacts and low specificity [5]. Magnetic resonance imaging (MRI) is known to provide highly sensitive breast images, but it is both costly and challenging to interpret accurately [5,6,7]. Positron emission tomography (PET) scans, although capable of early cancer detection, are limited by low spatial resolution [8]. Finally, while many family physicians perform Clinical Breast Examination (CBE) annually, its sensitivity ranges widely, from 40% to 69% [9]. Each imaging modality presents advantages and limitations, underscoring the necessity for complementary imaging techniques to provide comprehensive breast assessments in clinical practice [10].

In recent decades, advancements in optoelectronics and fiber optics have led to significant progress in near-infrared (NIR) optical imaging technologies, such as diffuse optical tomography (DOT) and diffuse optical spectroscopy (DOS) [2,11]. DOT, in particular, has demonstrated significant potential in providing functional insights crucial for identifying malignant tissues [12]. Its advantages include high sensitivity, non-invasiveness, absence of ionizing radiation, low cost, and suitability for repeated monitoring [13,14]. Several research teams have explored DOS and DOT probes to improve patient functionality and accuracy. One such squad at the University of California, Irvine, used single and multiple source–detector pairs in their DOS devices [15]. They investigated whether DOS measurements obtained before and 1 week into a 3-month neoadjuvant chemotherapy could predict the final post-surgical pathological response in 11 patients. Another group developed a device for breast spectroscopic imaging with a single three-wavelength LED at the center, surrounded by eight silicon diode detectors arranged in a circle around the source, with 4 cm separation [5]. They aimed to calculate angiogenesis and hypermetabolism effects on the cancerous breast compared to the corresponding normal tissue in the contralateral breast. Some other researchers, such as Vavadi et al., have employed other imaging techniques, like ultrasound, as a complementary method to improve the resolution and localization of their DOT device [16]. While these diffuse optical probes show promise in imaging highly scattering media like breast tissue [17], our team has pioneered the development of a patented CW-based handheld DOT probe intended specifically for breast cancer functional imaging [18,19].

Our overarching objective is to address the limitations of current breast cancer screening and monitoring devices by developing enhanced hardware for real-time imaging. Due to slow data acquisition and processing, many existing tools lack the speed and precision required for immediate clinical decisions. This paper presents an ongoing research project focused on designing a proprietary handheld diffuse optical breast-scanning probe for breast cancer detection, which we call the NIRscan probe, which overcomes these limitations throughout four development cycles, and our team has successfully refined a near-infrared handheld NIRscan probe for cross-sectional imaging, aimed at identifying potentially cancerous tissue through structural and functional analysis of biological tissues, based on variations in absorption and scattering within the near-infrared spectrum. For instance, different tissue components—such as water, fat, muscle, and blood—exhibit distinct characteristics in the NIR range [20].

The first version of the probe, developed in 2014 [7], employed encapsulated LEDs (eLEDs) with multi-wavelength, divergent-beam illumination instead of traditional laser-coupled fiber optics. This initial version was tested using optical phantoms and animal tissues to characterize tissue properties and refine imaging algorithms. The subsequent version, which incorporated significant updates from the previous design, was tested in clinical trials in 2016 at the Jim Pattison Outpatient Care and Surgery Centre (JPOCSC). These trials involved patients with known breast cancer and achieved an impressive accuracy rate of 95%, with a sensitivity of 0.947 and a specificity of 0.842 [21]. Despite these advancements, several limitations were identified, prompting us to enhance the current DOB-Scan probe by improving the CCD sensor, optimizing light intensity control, reducing noise, and streamlining the data collection process. Initially, the Farrel approximation [2] to the diffusion equation (DE) was employed to estimate reflectance values; however, the non-collimated nature of the light sources revealed several limitations in this model, restricting imaging accuracy [8,10,22].

Future developments will prioritize enhancing processing speed and positioning the NIRscan as a reliable, non-invasive solution for rapid breast cancer screening, monitoring, and diagnosis. This would impact clinical workflows by providing immediate feedback and enabling quicker and more accurate treatment planning.

### 1.1. Objectives/Goals Behind the Design of the Probe


*Functional images of breast tissue, allowing differentiation between healthy and cancerous tissues with high sensitivity;*

*Real-time imaging capabilities, providing immediate feedback for clinicians during examinations;*

*Portable and non-invasive form factor.*


### 1.2. Overview of the Probe

The NIRscan Probe is a handheld near-infrared optical imaging device focusing on real-time breast cancer detection and localization. Utilizing diffuse optical tomography (DOT), the probe can generate functional images of breast tissue by inferring key optical properties, such as scattering and absorption coefficients. This information enables differentiation between healthy and cancerous tissues, enhancing diagnostic accuracy. The device is equipped with a linear charge-coupled device (CCD) sensor and two encapsulated LEDs, which emit near-infrared light at specific wavelengths optimized for their penetration capabilities and sensitivity to different tissue components, such as oxyhemoglobin, deoxyhemoglobin, water, and fat [3].

Designed to be non-invasive, portable, and user-friendly, the NIR probe is suitable for use in bedside and clinical environments [1]. It incorporates an advanced image reconstruction algorithm to generate cross-sectional images of breast tissue, which aids in the precise localization and characterization of abnormalities [1]. As an improvement over traditional imaging modalities, the NIR probe offers enhanced sensitivity and specificity while avoiding the risks associated with ionizing radiation. Additionally, its real-time imaging capability provides immediate feedback for clinicians, thereby supporting faster and more informed clinical decision-making.

## 2. Materials and Methods

### 2.1. Details of Probe’s Hardware Design

#### 2.1.1. LED Light Sources

In NIR imaging, optimal wavelength selection is crucial for sufficient tissue penetration and high contrast between biological components. For this purpose, the probe utilizes two Al-GaAs LEDs (Marubeni America Corporation, Santa Clara, CA, USA; L690/750/800/850) [23] positioned 15 mm from each detector end, as depicted in Figure 1. Each LED includes four NIR wavelengths: 690, 750, 800, and 850 nm, chosen for their effective penetration through breast tissue and their differential response to different tissue components, such as oxyhemoglobin, deoxyhemoglobin, water, and fat [7]. The LEDs can be individually controlled, and the intensity for each wavelength is adjustable via an integrated LED driver IC (integrated circuit) located on the body PCB, enabling dynamic and precise light management during imaging.

Testing revealed several challenges, such as the nonlinear and inconsistent response of the light sources. Each LED had different current–power nonlinear characteristics, compounded by significant variation in LED manufacturing, leading to inconsistencies in power output between the eLEDs [8]. These variations necessitate rigorous calibration procedures to ensure consistent performance across multiple devices. Such inconsistencies complicated the normalization and processing of measured data.

The scattering nature of NIR light in biological tissues inherently limits the ability to achieve sufficient imaging resolution. As the light penetrates deeper, scattering increases, reducing the spatial resolution. This phenomenon is further exacerbated by heterogeneous tissue composition, making standardization across patients challenging. Therefore, balancing penetration depth and resolution is a key factor in the probe’s design, especially given the variability of breast tissue density, absorption, and thickness observed in clinical settings [10]. These factors must be considered for optimal performance in diverse patient conditions to achieve real-time imaging capabilities.

Moreover, precisely mounting LEDs and the CCD sensor is critical to ensure uniform light propagation and accurate detection. Any misalignment between these components can lead to uneven illumination and errors in the captured data, reducing the reliability of imaging results. Rigid and stable mounting enhances signal quality and minimizes the impact of external factors, such as vibrations or thermal expansion, which could disrupt the alignment during use. The system ensures consistency in illumination and detection by maintaining the LEDs and the CCD sensor in a flat and level configuration, ultimately improving imaging accuracy.

#### 2.1.2. Sensors

The NIRscan Probe utilizes a linear array CCD sensor, the Sony ILX511. It comprises 2048 active pixels, each with a 14 μm pitch, resulting in an effective imaging area of 28.672 mm. The sensor detects back-scattered photons. Since pixels closer to the light source are exposed to higher light intensity, the sensor’s design requires an extensive dynamic range to efficiently capture both the high and low intensity levels [24].

Matching the LED power output to the sensor’s detection capabilities is challenging, mainly due to the CCD sensor’s limited and narrow response band. The sensor has a spectral sensitivity of around 55% at 690 nm, which drops below 20% at 850 nm [23]. This reduced sensitivity affects the overall accuracy of NIR measurements and increases its susceptibility to visible ambient light, which can degrade the quality of data acquired during real-time imaging. To compensate for this variation in sensitivity, a wavelength-dependent calibration function needs to be developed to amplify the CCD sensor’s measurements with respect to its characteristic. The drawback of this method is the amplified noise on the frames measured from high-wavelength pulses.

Maintaining darkroom conditions to minimize ambient light may not always be feasible in clinical environments. To address this, an additional frame with all LEDs turned off is captured and subtracted from the subsequent frames, enhancing the signal quality. Achieving consistent imaging quality across different clinical settings remains challenging despite these measures.

#### 2.1.3. LED Driver

The LED light sources are powered by a TLC5916 8-channel constant-current LED sink driver [25], which allows for the precise control of each LED. Users can set intensity values via a graphical user interface (GUI), which are transmitted through USB to the microcontroller unit (MCU) to configure the LED driver. The relationship between user input and LED output power cannot be ignored, as it exhibited nonlinearity. Calibration procedures are essential to account for this nonlinearity and ensure expected linear optical output power, critical for imaging consistency in various settings.

#### 2.1.4. Analog-to-Digital Conversion

To convert the analog charge generated by back-scattered photons into digital data, an analog signal processor (AD9826) [26] is employed. It is programmed to function in 1-channel correlated double sampling (CDS) mode to sample the CCD array’s output. The maximum transmission rate possible (2 MHz, which is limited by manufacturing) restricts the data acquisition speed. However, real-time imaging necessitates minimizing processing delays. Although the current ADC setup supports a higher sampling rate, it still introduces challenges in achieving the required data rate for immediate imaging feedback without adding excessive high-frequency noise to the signal or losing data. A minimum transmission rate would be 10 MHz.

#### 2.1.5. Processing Unit

For real-time imaging, the digital data from the CCD sensor is processed by an ARM Cortex M4 (TM4C123GH, Texas Instruments, Dallas, TX, USA), an 80 MHz, 32-bit processor. This MCU controls the LEDs, CCD, and data acquisition. Given the need for immediate feedback, image averaging is performed to smooth the data and reduce computational load. However, real-time averaging on the MCU requires higher clock speeds than the current processor can support, which limits the system’s ability to provide instant analysis. As a workaround, averaging is implemented on the host computer, though this approach increases the computational load on the user’s system and may introduce lags in real-time applications. Future iterations will require an upgraded MCU to overcome these limitations and support the necessary real-time processing capabilities. See Figure 2.

### 2.2. Imaging Design and Reconstruction Algorithm

During imaging, the CCD sensor collects reflectance values in real time and transfers them to the host computer via USB 2.0. The software interface features a graphical user interface (GUI) that allows users to manually control the source wavelength and intensity, CCD integration time, and data collection [21]. See Figure 3.

Raw data from the CCD array are processed to reconstruct cross-sectional images of breast tissue. By averaging the reflectance from 16 contiguous pixels, the probe reduces noise and generates 128 data points, fitted to an analytical model to estimate the tissue’s optical properties. The final output includes cross-sectional images depicting the absorption parameters’ internal topology, allowing differentiation between healthy and cancerous tissue regions [27]. The advantages of this method in comparison to other numerical methods, such as finite element analysis or machine learning-based methods, are low computational cost, deterministic results, and the absence of a required training process.

Several studies have demonstrated the potential of the NIRscan probe for non-invasive breast cancer detection and monitoring, including imaging physical phantoms and clinical evaluations involving breast cancer patients. For example, Figure 4 shows a reconstructed 2D image of a physical phantom with a simulated abnormality. In contrast, Figure 5 illustrates the reconstructed 3D model of a patient’s tumor, highlighting the potential for precise tumor localization and characterization.

#### 2.2.1. Calculation of Absorption and Scattering Coefficients

The imaging process involves shining near-infrared light on the breast tissue using LEDs at various wavelengths, which are absorbed or scattered by the different tissue components. The CCD sensor collects the reflected light, and the measured light intensity provides information about the tissue’s properties. To determine the absorption and scattering coefficients, a mathematical model called the modified diffusion equation (MDE) is used, which was developed and refined in previous studies conducted by our lab [19,20]. This model helps us relate the measured light reflection to the tissue’s absorption (how much light is absorbed) and scattering (how much light bounces around inside the tissue). We can calculate these coefficients by adjusting the model until the predictions match the experimental data. However, changing the model to match the experimental measurements can be time-consuming and is not always automated. This adjustment process involves iteratively tweaking the parameters to achieve a fit, which requires significant computational resources and time, especially in a clinical setting. To overcome these limitations, machine learning (ML) and iterative numerical methods can be employed to automate and expedite this process. ML models trained on previous datasets can make the calculations more accurate, faster, and more consistent by reducing human intervention during model adjustments [8,28].

Additionally, these computational techniques can be expanded to construct 2D or 3D images of the tissue, providing a more comprehensive view than the simplified analytical models currently in use, which often assume a straightforward, single-path trajectory for the photons. ML and advanced algorithms allow for better visualization of complex light interactions in highly scattering biological tissues and enables a more detailed representation of inhomogeneities, crucial for precise tumor localization.

#### 2.2.2. Imaging Challenges

Biological tissues are generally highly turbid, meaning scatter interactions dominate absorption interactions. This leads photons to follow complex paths, and reflectance values collected by the CCD may be “contaminated” by light scattered from regions beyond the intended target. While stochastic models, like Monte Carlo simulations, could theoretically model this randomness, these methods are computationally demanding and unsuitable for real-time applications.

Integrating machine learning into imaging systems offers substantial potential for improving diagnostic accuracy and computational efficiency. Momtahen et al. demonstrated the application of machine learning in predicting residual cancer burden by processing optical properties with high precision. This approach not only automated complex calculations but also enhanced the speed and accuracy of the imaging process, achieving a high accuracy of 98% in predicting residual cancer burden (RCB) class [29]. Similarly, Shokoufi and Golnaraghi (2019) acknowledged the computational challenges inherent in traditional image reconstruction methods, particularly in handling scattering-induced complexities. They emphasized the need for advanced computational techniques to overcome these limitations and improve real-time imaging capabilities [7]. While our system currently does not employ AI, its integration in future iterations could significantly reduce manual intervention, improve data processing efficiency, and enhance imaging reliability. By enhancing the precision of absorption and scattering coefficient calculations, machine learning algorithms and iterative numerical methods can make the system even more robust in diverse clinical environments, enabling automated, precise, and efficient imaging processes well suited to clinical needs. See Figure 6.

## 3. Experimental Evaluation

The evaluation of the NIR probe can be divided into two categories: phantom tests and clinical trials. The former focused on the probe’s ability to calculate the optical properties of tissue-mimicking phantoms and reconstruct images, while the latter examined its practical use in a clinical setting.

### 3.1. Phantom Tests

Phantom tests were conducted using solid and liquid tissue-mimicking phantoms to simulate the optical properties of human breast tissue. These phantoms allowed for controlled assessment, where known optical properties could be compared with the measurements obtained from the probe. Delrin was chosen for solid phantoms due to its similar optical properties to human tissue, and Intralipid was used for liquid phantoms to represent fat tissue.

Reflectance and transmission tests were carried out with Delrin phantoms, using anomalies (such as iron inserts) to simulate the presence of tumors [19]. The data collected in these tests aimed to verify the probe’s accuracy in determining optical properties and distinguishing between healthy and anomalous tissue regions. Liquid phantom experiments evaluated the probe’s sensitivity to different chromophore concentrations, using varying concentrations of Intralipid as the diffusive medium and solid iron as an absorbing anomaly [19]. The results indicated variability in reflectance and transmission patterns, suggesting challenges in measurement consistency, possibly due to scattering artifacts and sensor calibration issues.

### 3.2. Clinical Trials

The clinical trial phase of this study involved testing the probe on patients undergoing neoadjuvant chemotherapy (NAC). Optical measurements were taken before and after each cycle of NAC, and the probe’s ability to monitor changes to predict the residual cancer burden (RCB) was assessed [2]. Results were favorable, with the NIR Probe providing high accuracy in predicting treatment outcomes. Further research is being conducted on this dataset. As mentioned, a machine learning-based approach processes the data collected, incorporating the MDE.

### 3.3. Experimental Findings

The experiments demonstrated strengths and limitations in the current design of the NIR Probe. The phantom tests indicated that while the probe could detect anomalies, the inconsistency in measured absorption and scattering coefficients highlighted the need to further optimize sensor calibration and image reconstruction methods [2]. Clinical trials integrating machine learning models show the utility of the NIR Probe in clinical decision-making concerning the effectiveness of chemotherapy as determined by RCB. Still, additional improvements in hardware components, such as LED power control and CCD sensitivity, will be necessary to achieve consistent performance in real-world clinical applications and reduce measurement noise [20].

## 4. Next Steps and Conclusion

The NIRscan probe has demonstrated promising real-time breast cancer imaging and detection capabilities. However, various challenges were identified during the experimental evaluations, highlighting the need for several hardware and software component improvements to meet the initial goals of providing a portable, non-invasive imaging tool capable of real-time clinician feedback.

Next Steps for Improvement:
*Enhanced Signal Processing*○Upgrade the microcontroller unit (MCU) to a more robust model to enable real-time signal noise filtering. This enhancement would significantly reduce the computational load on the host computer, ensuring efficient, real-time imaging.


*Calibration and Control Optimization*
○Implement a more robust calibration method for the LEDs and CCD sensor to address inconsistencies in light intensity and improve reflectance measurements. This is important to ensure that data from various clinical settings are usable.



*Integration of Machine Learning*
○Machine learning algorithms that effectively differentiate scattered light from inhomogeneities from healthy tissue must be further developed and incorporated to improve image reconstruction. These algorithms can enhance the precision of absorption and scattering coefficient calculations by effectively differentiating them from inhomogeneities.



*Algorithm Improvement*
○Refine the reconstruction algorithm to minimize dependency on initial guesses for absorption and scattering coefficients. The reliance on manual adjustments contributes to variability in imaging results, which could be reduced by utilizing dynamic machine learning models that automatically adjust initial parameters.



*Sensor Upgrades*
○The existing CCD sensor should be replaced with an advanced NIR-optimized sensor to increase sensitivity, particularly in the 850 nm range. This upgrade would improve detection accuracy for deeper tissue penetration, resulting in more reliable imaging outcomes.



*Mechanical Design Adjustments*
○Improve the physical design to ensure consistent placement of LEDs and sensors on the PCB. This adjustment will help reduce variability in LED orientation, minimize errors in light propagation, and improve overall measurement reliability in clinical environments.



*Real-Time Processing Capabilities*
○Add a dedicated digital signal processing (DSP) unit to calculate real-time absorption and scattering coefficients. This enhancement will empower the probe to provide immediate, actionable feedback during clinical evaluations.


The ultimate objective of these improvements is to provide a reliable, portable, and non-invasive breast cancer imaging tool. The next iteration’s key focus is addressing the hardware limitations that currently inhibit real-time capabilities and improving the accuracy and consistency of image reconstruction.

## 5. Conclusions

The NIRscan probe demonstrates significant potential as a tool for breast cancer detection and localization. It combines near-infrared diffuse optical tomography (DOT) with modern machine learning techniques to support non-invasive imaging. Despite challenges related to hardware limitations and algorithmic dependencies, this device offers a pathway to a more advanced breast cancer screening method.

However, for successful clinical adoption, several obstacles must be addressed. These include variability in breast tissue density and optical properties among patients, the impact of ambient light on image quality in non-controlled clinical environments, and the need for hardware improvements, particularly in sensor sensitivity at higher wavelengths. Additionally, the probe’s reliance on real-time computational resources and machine learning algorithms necessitates high-performance hardware, which may limit its use in resource-constrained settings. Practical clinician training and adherence to regulatory standards will also be critical to ensure safe and efficient implementation.

Future improvements will focus on overcoming these obstacles by enhancing real-time imaging capabilities, optimizing signal processing algorithms, and increasing diagnostic accuracy. By addressing these challenges, the next version of the probe aims to establish itself as a more capable and efficient breast cancer detection and monitoring tool, ultimately providing clinicians with valuable, timely insights for effective decision-making in patient care.

## Figures and Tables

**Figure 1 sensors-25-00983-f001:**
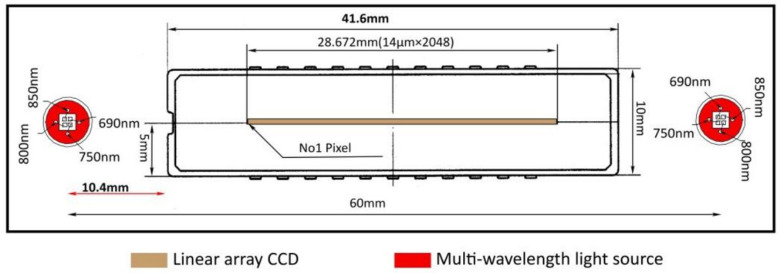
Face of probe head [1].

**Figure 2 sensors-25-00983-f002:**
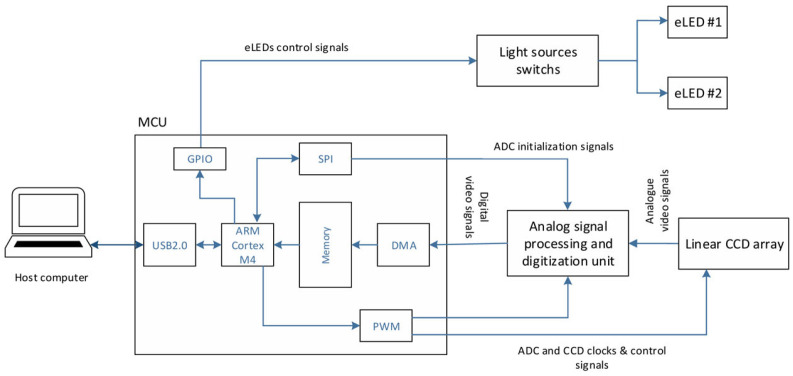
System architecture of the NIR handheld probe, showing the integration of the ARM Cortex M4 microcontroller, ADC, CCD sensor, and light source control [1].

**Figure 3 sensors-25-00983-f003:**
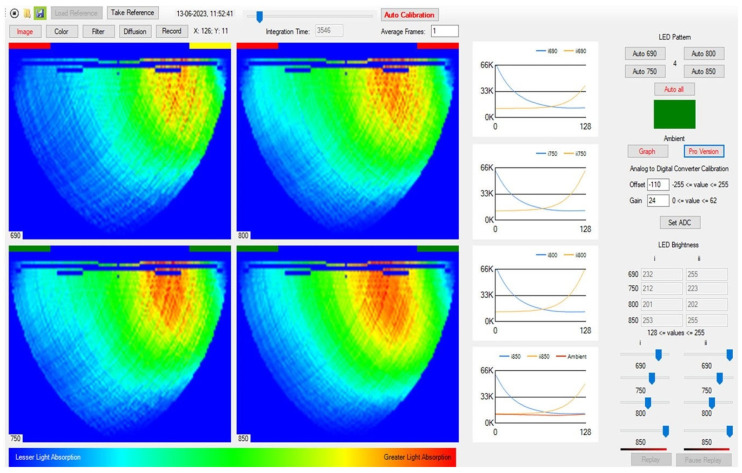
The custom software’s graphical user interface (GUI) version 0.4 is presented on a Windows platform.

**Figure 4 sensors-25-00983-f004:**
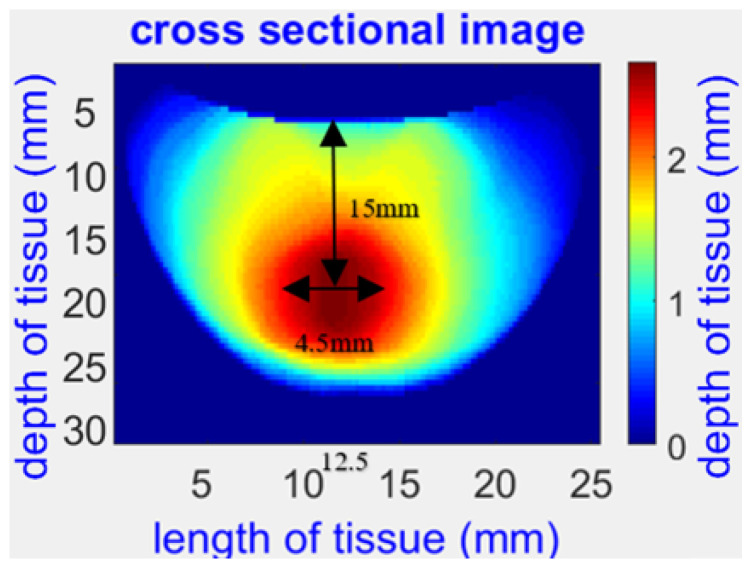
An optical image of a physical phantom with a 4.5 mm spherical abnormality at the center was captured at 690 nm [1].

**Figure 5 sensors-25-00983-f005:**
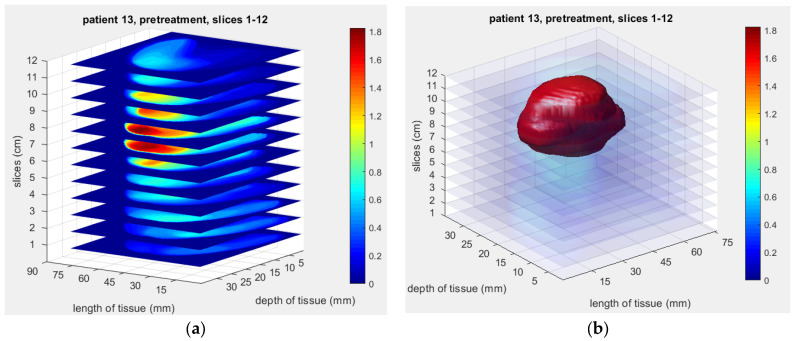
Reconstructed 3D optical image of a patient’s tumor using 12 slices at 690 nm: (**a**) Reconstructed image using the NIR probe’s MDE imaging. (**b**) A 3D volume model of the tumor created using MATLAB rendering capabilities with gaps between the adjacent slices interpolated [19].

**Figure 6 sensors-25-00983-f006:**
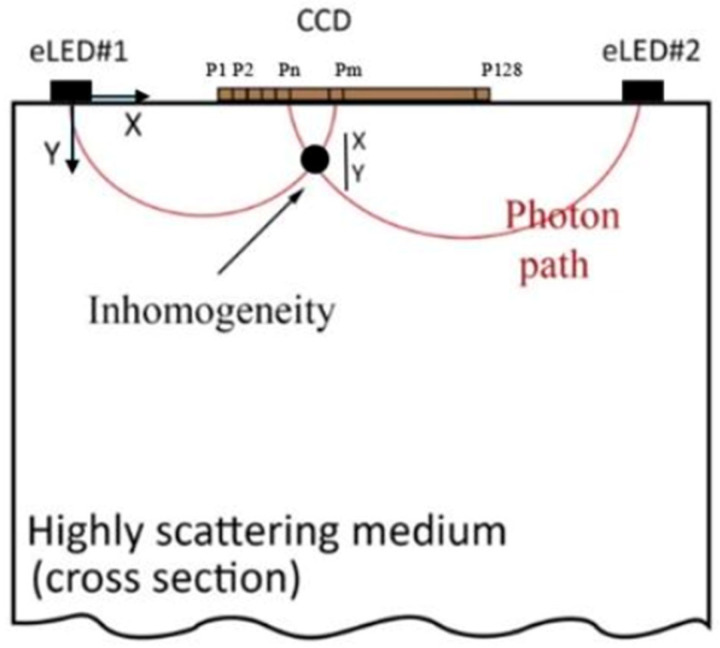
Diagram illustrating the photon propagation paths in a highly scattering medium. LED light travels along multiple, semi-circular paths through the tissue, converging at various points [19].

## Data Availability

Data are contained within the article.

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
