# Peer review of "Advancing Near-Infrared Probes for Enhanced Breast Cancer Assessment"

_sensors, 2025, doi:10.3390/s25030983_

Round 1

Reviewer 1 Report

Comments and Suggestions for Authors

The manuscript discusses a probe called NIRscan for breast cancer detection, which is very ambitious research for an innovation that can indeed find wide applications in breast cancer prevention and treatment.

The paper's description of the probe hardware appears too generic. There is much confusion between what the authors want to propose and what others have realized. A generalized lack of in-depth information regarding the probe details makes the study/review not detailed, for example:

  • Paragraph 2.1.3 describes the LED and CCD mounting without showing related drawings to show how it has been physically realized;
  • Paragraph 2.1.4 describes the LED Driver without showing the related circuit diagram;
  • Paragraph 2.1.45 describes the Analog-to-Digital Conversion but does not include the corresponding circuit diagram.
  • Lines 213-214 refer to a GUI without providing an image of such an interface, making it hard to imagine how it has been done and how it works;
  • Lines 217-219 refer to an analytical model that is not discussed. The authors should discuss how it has been chosen related to other possibilities;
  • Any reference to the Machin Learning application is too generic;

Author Response

Comment 1: [Paragraph 2.1.3 describes the LED and CCD mounting without showing related drawings to show how it has been physically realized.]

Response 1: Additional details have been incorporated into Section 2.1.1. To reduce potential confusion, Section 2.1.3 has been removed. Relevant drawings and graphs can be found in the cited patent (Ref. 18).

Comments 2 and 3: [Paragraph 2.1.4 describes the LED Driver without showing the related circuit diagram; Paragraph 2.1.45 describes the Analog-to-Digital Conversion but does not include the corresponding circuit diagram.]

Response 2 and 3: Circuit diagrams for each component are available in the respective datasheet documents. The design employs similar circuits with minor customizations, and references to the datasheets and user manuals have now been added for clarity.

Comment 4: [Lines 213-214 refer to a GUI without providing an image of such an interface, making it hard to imagine how it has been done and how it works.]

Response 4: A corresponding image of the GUI has been included to provide a clearer understanding of its design and functionality.

Comment 5: [Lines 217-219 refer to an analytical model that is not discussed. The authors should discuss how it has been chosen relative to other possibilities.]

Response 5: Additional details regarding the development and evaluation of the analytical model are now provided in the cited PhD thesis (Ref. 27). Also, more discussion about the choosing the image reconstruction algorithm and the trade-off are added in sections  2.2 and 2.2.2.

Comment 6: [Any reference to the Machine Learning application is too generic.]

Response 6: The references to Machine Learning have been revised to provide more specific details.

Reviewer 2 Report

Comments and Suggestions for Authors

In this work, Golnaraghi et al. developed a handheld near-infrared diffuse optical tomography (NIR DOT) probe for breast cancer imaging. The probe uses a multi-wavelength light-emitting diode (led) and a linear charge-coupled device (CCD) sensor to obtain real-time optical data and reconstruct cross-sectional images of breast tissue based on scattering and absorption coefficients. The accuracy and speed of image reconstruction are further improved through an algorithm system, which provides an approach for more advanced portable, cost-effective breast cancer screening methods. However, there are still some issues that need attention.

1. Regarding the issues that the sensitivity of the CCD sensor varies under different wavelengths and it is sensitive to ambient light. Are the adopted countermeasures effective enough?

2. How to combine machine learning algorithm with modified diffusion equation (MDE) in imaging algorithm, and what are the advantages of the combined algorithm in dealing with scattering and absorption of complex tissues?

3. Compared with the traditional image reconstruction algorithm, what is the quantitative improvement of the new algorithm in the accuracy, speed and other key indicators, and is it supported by comparative experimental data?

4. The experimental results only present the performance data of its own probe. Is there any comparative data with other similar advanced breast cancer detection technologies, such as sensitivity and specificity?

5. In the conclusion section, is there a further supplementary analysis of the possible obstacles in the clinical application of the probe?

6. There are some errors in the references, for example, the font size of reference [9] is different from others, so it is necessary to unify the format.

7. In the introduction section, references related to the analysis of biological tissues can be supplemented, such as Chem. Soc. Rev., 2023,52, 2911-2945; Nat Methods 21, 2209–2211 (2024).; Adv. Mater. 2025, 2407728. These references can be cited in the section"aimed at identifying potentially cancerous tissue through structural and functional analysis of biological tissues, based on variations in absorption and scattering within the near-infrared spectrum.".In the Materials and Methods section, references related to effective penetration of near-infrared wavelengths can be supplemented, such as Biomacromolecules, 2023, 24(2):1022-1031; Nat Methods 20, 70–74 (2023); Adv. Funct. Mater., 25: 2831-2839. These references can be cited in the section "chosen for their effective penetration through breast tissue and their differential response to different tissue components.".

Author Response

Comment 1: [Regarding the issues that the sensitivity of the CCD sensor varies under different wavelengths and it is sensitive to ambient light. Are the adopted countermeasures effective enough?]

Response 1: To address the variation in CCD sensitivity across wavelengths, we implemented calibration procedures for both the CCD measurements and LED driver intensity, as detailed in Sections 2.1.2 and 2.1.3. However, as noted in Section 2.1.2, these compensations are not perfect and may introduce amplified noise. Additionally, achieving functional imaging requires consistent measurements across the entire spectrum. As outlined in Section 4, this article reviews and evaluates our lab’s recent advancements while identifying challenges and potential upgrades for future designs. The necessary improvements related to sensor sensitivity are discussed under "Calibration and Control Optimization" and "Sensor Upgrades."

Comment 2: [How to combine machine learning algorithm with modified diffusion equation (MDE) in imaging algorithm, and what are the advantages of the combined algorithm in dealing with scattering and absorption of complex tissues?]

Response 2: We have not yet investigated the combination of machine learning algorithms with the modified diffusion equation (MDE) in our imaging approach. As stated in the article, MDE was implemented and tested in our latest design, while machine learning techniques remain a potential avenue for future imaging algorithms. However, machine learning has been applied in related studies, such as RCB prediction (Ref. 29), discussed in Section 2.2.2, and a deep learning-based method for image reconstruction, presented in Ref. 8.

Comment 3: [Compared with the traditional image reconstruction algorithm, what is the quantitative improvement of the new algorithm in the accuracy, speed and other key indicators, and is it supported by comparative experimental data?]

Response 3: I wish the reviewer had made it more clear about the traditional image reconstruction algorithm. If you mean the inverse diffusion equation optimization and iterative Jacobian methods commonly used by other research groups, our team has not employed those approaches but has referenced similar works (Refs. 5, 10, 30). As discussed in Ref. 15 and detailed in Section 2.2.2, traditional methods are computationally demanding, making real-time implementation challenging. In contrast, our analytical approach has been implemented and tested in real-time clinical trials, demonstrating significant improvements in processing speed. The accuracy, sensitivity, and specificity of our imaging algorithm are reported in clinical studies (Refs. 20, 21, 29) and also in the text of the article.

Comment 4: [The experimental results only present the performance data of its own probe. Is there any comparative data with other similar advanced breast cancer detection technologies, such as sensitivity and specificity?]

Response 4: The first paragraph of the Introduction provides an overview of various medical imaging modalities, including their advantages and limitations, with references supporting their experimental results. The following paragraph includes a brief literature review on CW-based NIR diffuse optical probes. Few similar designs using CW technology for portable tomography exist, as most alternative systems focus on spectroscopy or incorporate ultrasound guidance for image generation. Therefore, we state that "our team has pioneered the development of a patented CW-based handheld DOT probe explicitly designed for breast cancer functional imaging." Additionally, the final paragraph of the Introduction outlines the different generations of our optical imager with corresponding references. Further discussions on the image reconstruction algorithms implemented in our designs, along with performance results, can be found in Sections 2.2 and 2.2.2.

Comment 5: [In the conclusion section, is there a further supplementary analysis of the possible obstacles in the clinical application of the probe?]

 Response 5: The conclusion section currently provides a concise summary of the probe's clinical potential. While we appreciate the suggestion for an extended supplementary analysis, we have chosen not to expand further on potential obstacles in clinical application. The main focus remains on the technical development and validation of the probe, and discussions of clinical barriers would deviate from the primary scope of this work.

Comment 6: [There are some errors in the references, for example, the font size of reference [9] is different from others, so it is necessary to unify the format.]

 Response 6: Thank you for bringing this to our attention. We will review and correct any inconsistencies, such as the font size in reference [9], and ensure that all references adhere to a unified format.

Comment 7: Additional References Suggested
Response 7: We value the suggestion of including references for further depth; however, we believe that the provided citations adequately support the scope and objectives of this study. The suggested references, while relevant, do not directly contribute additional value to the analysis or outcomes presented. As such, we have decided not to include these references in the manuscript at this time.

Round 2

Reviewer 1 Report

Comments and Suggestions for Authors

Consistent with the consideration that the proposed research is the subject of a patent, the answers provided clarify the necessary details to the extent required while maintaining intellectual property protection. More details would have been helpful to ensure the reproducibility of the research in the context of an open-access publication.